# An Improved VMD-LSTM Model for Time-Varying GNSS Time Series Prediction with Temporally Correlated Noise

Hongkang Chen [1], Tieding Lu [1], Jiahui Huang [2], Xiaoxing He [2,*], Kegen Yu [3], Xiwen Sun [1], Xiaping Ma [4] and Zhengkai Huang [5]

1　School of Geodesy and Geomatics, East China University of Technology, Nanchang 341000, China; chk@ecut.edu.cn (H.C.); tdlu@ecut.edu.cn (T.L.); xwsun@ecut.edu.cn (X.S.)
2　School of Civil and Surveying & Mapping Engineering, Jiangxi University of Science and Technology, Ganzhou 341000, China; hjh@mail.jxust.edu.cn
3　School of Environment and Surveying, China University of Mining and Technology, Xuzhou 221000, China; kegen.yu@cumt.edu.cn
4　School of Surveying and Mapping Science and Technology, Xi'an University of Science and Technology, Xi'an 710000, China; celiang0321@163.com
5　School of Transportation Engineering, East China Jiao Tong University, Nanchang 330013, China; zhkhuang@whu.edu.cn
*　Correspondence: xxh@jxust.edu.cn

**Abstract:** GNSS time series prediction plays a significant role in monitoring crustal plate motion, landslide detection, and the maintenance of the global coordinate framework. Long short-term memory (LSTM) is a deep learning model that has been widely applied in the field of high-precision time series prediction and is often combined with Variational Mode Decomposition (VMD) to form the VMD-LSTM hybrid model. To further improve the prediction accuracy of the VMD-LSTM model, this paper proposes a dual variational modal decomposition long short-term memory (DVMD-LSTM) model to effectively handle noise in GNSS time series prediction. This model extracts fluctuation features from the residual terms obtained after VMD decomposition to reduce the prediction errors associated with residual terms in the VMD-LSTM model. Daily E, N, and U coordinate data recorded at multiple GNSS stations between 2000 and 2022 were used to validate the performance of the proposed DVMD-LSTM model. The experimental results demonstrate that, compared to the VMD-LSTM model, the DVMD-LSTM model achieves significant improvements in prediction performance across all measurement stations. The average *RMSE* is reduced by 9.86% and the average *MAE* is reduced by 9.44%; moreover, the average *R2* increased by 17.97%. Furthermore, the average accuracy of the optimal noise model for the predicted results is improved by 36.50%, and the average velocity accuracy of the predicted results is enhanced by 33.02%. These findings collectively attest to the superior predictive capabilities of the DVMD-LSTM model, thereby demonstrating the reliability of the predicted results.

**Keywords:** GNSS; deep learning; time series prediction; VMD; LSTM

## 1. Introduction

Over the past three decades, with the rapid development of satellite navigation technology, many GNSS continuously operating reference stations have been established worldwide. These stations provide important data sources for crustal plate motion monitoring [1–5], landslide detection [6–8], the deformation monitoring of bridges or dams [9–13], and the maintenance of regional or global coordinate frameworks [14,15]. By analyzing the long-term GNSS observation data time series obtained from these stations, it is possible to predict the variation of coordinates at continuous time points, thereby providing an important basis for determining motion trends. This has significant practical and theoretical value in geodesy and geodynamics research [16–18].

Time series prediction methods can be mainly categorized into two types: physical simulation and numerical simulation [19,20]. Traditional physical and numerical simulation methods rely on geophysical theories, linear terms, periodic terms, and gap information to construct models [21]. However, these models face challenges in capturing complex nonlinear data and require a manual selection of feature information and modeling parameters, leading to systematic biases and limitations [22]. In contrast, deep learning, as an emerging technology, can automatically extract information that is suitable for data features by constructing deep network structures. Deep learning exhibits strong learning capabilities and has advantages in handling large-scale and high-dimensional data. It has been widely applied in various fields such as image recognition [23–25], natural language processing [26–28], speech recognition [29–31], and time series prediction [32–36]. Li et al. (2022) comprehensively analyzed and elaborated on the application of image recognition to plant phenotypes by comparing and analyzing various deep learning methods [24]. Otter et al. (2020) summarized and analyzed the relevant research of deep learning models in the field of natural language processing and provided valuable suggestions for future research in this field [26]. Nassif et al. (2019) systematically studied its accuracy in speech recognition through convolutional, recurrent, and fully connected deep learning methods [31]. Masini et al. (2023) elaborated on the application of machine learning in the field of economy and finance by analyzing the application of different neural networks and tree structures in time series in the context of deep learning [36].

Long short-term memory (LSTM), as an excellent variant of recurrent neural networks (RNNs), overcomes the issues of gradient vanishing, gradient exploding, and insufficient long-term memory in RNNs [37–39]. Due to its significant advantages in long-range time series prediction, LSTM has been widely applied in various time series prediction domains such as electricity load forecasting [40–42] and wind speed prediction [43–45]. In recent years, the application of the LSTM algorithm in the GNSS domain has also become increasingly widespread. Kim et al. (2019) improved the accuracy and stability of absolute positioning solutions in autonomous vehicle navigation using a multi-layer LSTM model [46]. Tao et al. (2021) utilized a CNN-LSTM approach to extract deep multipath features from GNSS coordinate sequences, reducing the impact of multipath effects on positioning accuracy [47]. Xie et al. (2019) accurately predicted landslide periodic components using the LSTM model to establish a landslide hazard warning system [48].

Variational Mode Decomposition (VMD) is a signal processing method based on the principle of variational inference. It decomposes signals into various mode components (Intrinsic Mode Functions, IMF) with different frequencies through an optimization process, effectively extracting the local time–frequency features of signals and enabling efficient signal decomposition and analysis [49–51]. Currently, many researchers have combined VMD with LSTM to enhance the performance of LSTM in a range of fields [52–55]. Huang et al. (2022) applied the VMD-LSTM model in the coal seam thickness prediction field, confirming that the predicted results closely matched the coal seam information obtained from existing boreholes [56]. Zhang et al. (2022) applied the VMD-LSTM model in the field of sports artificial intelligence, demonstrating its broad application prospects in predicting sports artificial intelligence directions [57]. Han et al. (2019) applied the VMD-LSTM model in the wind power prediction field, validating its high performance in multi-step and real-time predictions [58]. Xing et al. (2019) applied the VMD-LSTM model in predicting the dynamic displacements of landslides and verified its high prediction accuracy using the case of landslides in paddy fields in China [59].

The VMD-LSTM model has been widely adopted in various fields for time series prediction. However, most studies utilize VMD to decompose the original data, predict each Intrinsic Mode Function (IMF) and residual term separately, and then combine the predicted results to obtain the final prediction. Although this method yields good results for each IMF value, the fluctuation characteristics of the residual term are difficult to extract, leading to significant prediction errors in the model. Furthermore, existing research has mainly focused on the accuracy of the prediction results while neglecting the noise

characteristics of the data itself [60–62]. Considering these factors, this paper proposes a dual VMD-LSTM (DVMD-LSTM) hybrid model that considers the characteristics of noise. By performing VMD decomposition on the residual components obtained from the initial VMD decomposition, the proposed model effectively extracts the fluctuation features within the residuals, enabling the high-precision prediction of GNSS time series. By analyzing the *RMSE* and *MAE* and *R2* (coefficient of determination) of the predicted results in the E, N, and U directions across multiple measurement stations, the applicability and robustness of the proposed method are evaluated. Additionally, the quality of the predicted results is assessed by incorporating noise models and velocity evaluation.

The structure of this paper is as follows: Section 2 introduces the principles of VMD, LSTM algorithms, and accuracy evaluation metrics. The principles and specific processes of the DVMD-LSTM model are explained in detail. Section 3 describes the GNSS station data, presents data-preprocessing strategies, and analyzes reasons for the improved accuracy of the DVMD-LSTM model. Section 4 focuses on the prediction results and accuracy of both the single LSTM model and the hybrid model. The optimal noise model and velocity under each prediction model are compared and analyzed to evaluate the performance of the DVMD-LSTM model using different accuracy assessment metrics. Finally, Section 5 provides conclusions and an analysis.

## 2. Principle and Method

### 2.1. Variational Modal Decomposition (VMD)

Variational Mode Decomposition (VMD) is an adaptive and fully non-recursive method used for solving modal variational and signal processing problems [63]. GNSS time series exhibit inherent non-stationarity. Utilizing VMD to decompose the data effectively separates it into stationary signals, thereby extracting the fluctuation characteristics of the GNSS time series and providing a superior data foundation for model prediction. VMD iteratively searches for a variational model to decompose the original time series into distinct modal components. The specific decomposition process is outlined as follows [64–66]:

(1) For each modal component $\mu_K(t)$, the corresponding analytic signal is computed using the Hilbert transform, which allows its one-sided spectrum to be obtained:

$$\left[ \delta(t) + \frac{j}{\pi t} \right] * \mu_K(t) \tag{1}$$

In the equation, $j^2 = -1$, where $\delta$ is the Dirac distribution.

(2) By introducing exponential terms in each mode, the center frequency $e^{-j\omega_K t}$ of each mode can be estimated, and the spectral components of each mode can be modulated to their respective fundamental frequency bands:

$$\left[ \left( \delta(t) + \frac{j}{\pi t} \right) * \mu_K(t) \right] e^{-j\omega_K t} \tag{2}$$

(3) The bandwidth of $\omega_K$ is estimated based on the smoothness of the demodulated signal's H1 Gaussian. This leads to a constrained variational problem:

$$\min_{\{\mu_K\}, \{\omega_K\}} \left\{ \sum_K \| d_t \left[ \left( \delta(t) + \frac{j}{\pi t} \right) * u_K(t) \right] e^{-j\omega_K t} \|_2^2 \right\} \tag{3}$$

$$s, t, \sum_K \mu_K = f \tag{4}$$

In the equation, $f$ represents the original signal, $\{\mu_K\}$ represents the decomposed mode functions, and $\{\omega_K\}$ represents the corresponding center frequencies of each mode.

(4) On this basis, quadratic penalty factors $\alpha$ and the Lagrange multiplier operator $\lambda_t$ are introduced to transform it into an unconstrained variational problem. The extended Lagrange expression is as follows:

$$L(\{\mu_K\},\{\omega_K\},\lambda) = \alpha\sum_K \left\| \partial_t \left[ \left(\delta(t) + \frac{j}{\pi t}\right) * \mu_K(t)\right]e^{-j\omega_K t}\right\|_2^2 + \left\| f(t) - \sum_K \mu_K(t)\right\|_2^2 + \left\langle \lambda(t), f(t) - \sum_K \mu_K(t)\right\rangle \quad (5)$$

where $\alpha$ represents the quadratic penalty factor and $\lambda_t$ denotes the Lagrange multiplier operator. Subsequently, the alternating direction method of multipliers (ADMMs) is employed to solve this unconstrained variational problem. By alternately updating $\mu_K{}^{n+1}$, $\omega_K{}^{n+1}$, and $\lambda^{n+1}$, the saddle point of the extended Lagrange expression, i.e., the optimal solution of the constrained variational model in Equation (3), is sought.

### 2.2. Long Short-Term Memory (LSTM)

LSTM is an improved type of recurrent neural network (RNN) that addresses the issue of long-term dependencies by utilizing memory cells, effectively mitigating the problems of vanishing and exploding gradients [67–69]. Compared to traditional neural networks, LSTM demonstrates strong advantages in handling long-term sequence prediction tasks and has been widely applied in areas such as time series forecasting and fault detection [70–74]. The LSTM architecture consists of input layers, hidden layers, and output layers, where each hidden layer employs input gates, forget gates, and output gates to store and access data, as shown in Figure 1.

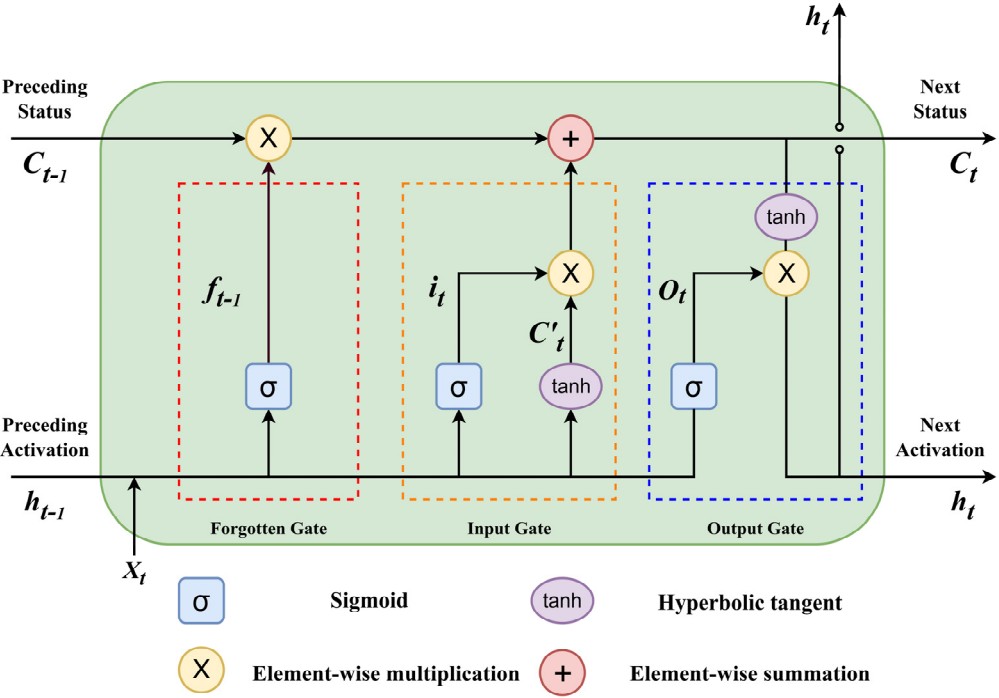

**Figure 1.** Basic structure of LSTM.

### 2.3. Dual Variational Mode Decomposition Long-Short Term Memory Network Model (DVMD-LSTM)

The VMD-LSTM model, as a classical hybrid deep learning model, has been widely applied in time series prediction tasks such as load forecasting and wind speed prediction, demonstrating remarkable predictive accuracy [75,76]. This model utilizes the Variational Mode Decomposition (VMD) to decompose the original data into a set of Intrinsic Mode Functions (IMFs) and a residue component, denoted as "r.". Subsequently, each IMF and the residue component are individually predicted, and their predictions are cumulatively aggregated to obtain the final model's prediction. It is worth noting that the IMFs, being

stationary signals, can achieve higher predictive accuracy when they are individually predicted; thus, effectively enhancing the predictive performance of the VMD-LSTM model. The specific prediction process is shown on the left side of Figure 2, and the residual value is not decomposed. However, the residue component remains unprocessed during the prediction process, leading to errors that can affect the model's predictive accuracy. Considering that the residual terms obtained after the VMD decomposition of real-world data still exhibit certain fluctuation characteristics and non-white noise such as high-frequency noise [77,78], this model further decomposes the residual terms using VMD and predicts the decomposed mode components to mitigate the impact of incomplete VMD decomposition. The DVMD-LSTM model improves the overall prediction accuracy by replacing the predicted results of the original residual terms with the fused mode components, thereby reducing the influence of residual terms on the prediction accuracy. The specific workflow is illustrated in Figure 2.

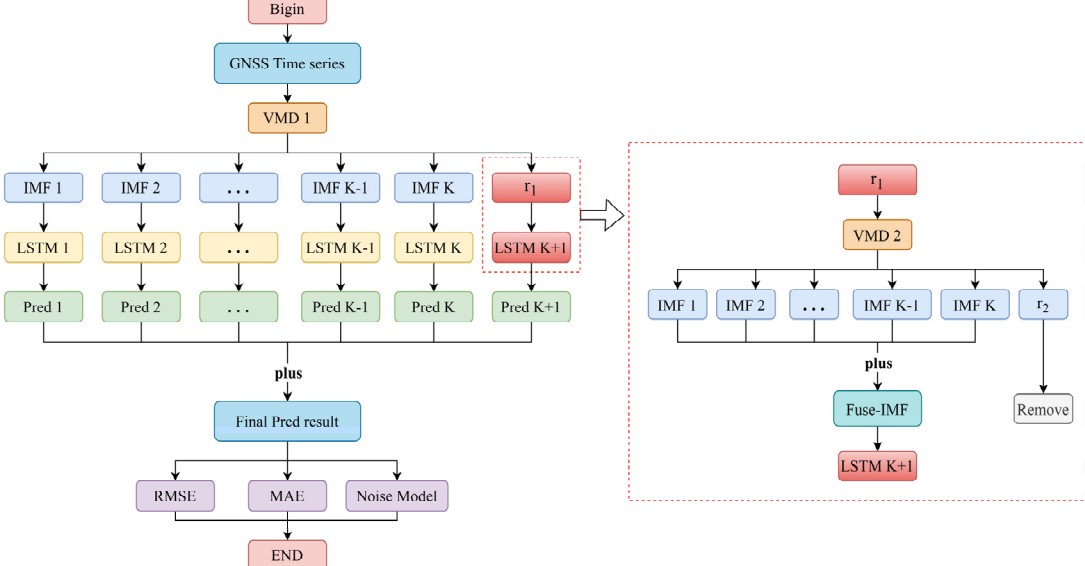

**Figure 2.** DVMD-LSTM hybrid model prediction process.

The specific prediction process of the DVMD-LSTM model is as follows:

Step 1: Preprocess the GNSS time series data by removing outliers, performing interpolation, and other data preprocessing techniques. Then, input the preprocessed data into the Variational Mode Decomposition (VMD) for decomposition.

Step 2: Further decompose the residue component "$r_1$" obtained from the VMD into individual modal components and another residue "$r_2$" through another round of VMD.

Step 3: Add up the modal components obtained from the VMD decomposition of the residue component "$r_1$" to form the fused Intrinsic Mode Function (Fuse-IMF). Use the Fuse-IMF as a feature for prediction in the LSTM model.

Step 4: Use the individual modal components obtained from the VMD decomposition of the original GNSS time series as features and input them separately into the LSTM model for prediction. Obtain $K$ prediction results, where $K$ represents the number of modal components.

Step 5: Add the $K$ prediction results obtained in Step 4 with the prediction result of the Fuse-IMF to obtain the final prediction result of the DVMD-LSTM model.

Step 6: Calculate the *RMSE* and *MAE* of the prediction results and use them to evaluate the performance of the model under different noise models.

### 2.4. Precision Evaluation Index

To evaluate the prediction accuracy and noise characteristics of the hybrid model, this study employs Root Mean Square Error (*RMSE*), Mean Absolute Percentage Error

(MAPE), and coefficient of determination (*R2*) as evaluation metrics for model prediction accuracy [79,80]. Additionally, the Bayesian information criterion (*BIC_tp*) is used to determine the optimal noise model for the original GNSS time series and the predicted time series under each model to determine whether the prediction results consider colored noise [81–83]. The definitions of the three evaluation metrics are as follows:

(1) *RMSE*

$$RMSE = \sqrt{\frac{1}{n} \sum_{i=1}^{n} (y_i - \hat{y}_i)^2} \tag{6}$$

(2) *MAE*

$$MAE = \frac{1}{n} \sum_{i=1}^{n} |(y_i - \hat{y}_i)| \tag{7}$$

(3) *R2*

$$R2 = 1 - \frac{\sum\limits_{i=1}^{n} (y_i - \hat{y}_i)^2}{\sum\limits_{i=1}^{n} (y_i - \overline{y})^2} \tag{8}$$

In the above equations, $y_i$ represents the actual GNSS data values, $\overline{y}$ represents the mean of actual GNSS data values, $\hat{y}_i$ represents the predicted results of each model, and $n$ denotes the number of GNSS data points. The values of *RMSE* and *MAE* are used as evaluation metrics for model prediction accuracy. Smaller values of *RMSE* and *MAE* indicate the higher prediction accuracy of the model, while larger values indicate a lower prediction accuracy. The coefficient of determination (*R2*) ranges between 0 and 1. When *R2* is close to 1, it indicates that the prediction model can explain the variability of the dependent variable well. On the other hand, when *R2* is close to 0, it suggests that the explanatory power of the prediction model is weak.

(4) *BIC_tp*

$$BIC\_tp = -2\log(L) + \log(\frac{n}{2\pi})v \tag{9}$$

To provide a visual assessment of the improvement achieved by the hybrid model on each evaluation metric, this study introduces the Improvement Ratio (*I*) to quantify the magnitude of improvement in each accuracy evaluation metric. By calculating the I value, the degree of improvement in accuracy achieved by the hybrid model can be accurately determined. The calculation formula for the Improvement Ratio is as follows:

$$I_{y\hat{y}} = \frac{y - \hat{y}}{y} \tag{10}$$

In the above equation, $y$ and $\hat{y}$ represent the evaluation metrics for accuracy, such as *RMSE*. The variable $y$ represents the evaluation metric for the accuracy of the initial model's predictions, while $\hat{y}$ represents the evaluation metric for the accuracy of the predictions made by the hybrid model. A larger value of $I_{y\hat{y}}$ indicates a greater improvement in the evaluation metric achieved by the hybrid model and vice versa.

## 3. Data and Experiments

### 3.1. Data Sources

In this work, the daily time series of 8 GNSS stations (ENU) from the Extended Solid Earth Science ESDR System (ES3) were selected for the experiment. The GNSS daily loose constraints solution from GAMIT and GIPSY was used with Quasi-Observation Combination Analysis (QOCA) software to generate a combined solution [62]. The information for each station is presented in Table 1, and the distribution of the stations is depicted in Figure 3. See Appendix A for details of data fluctuations.

**Table 1.** Information of each GNSS station.

| Site | Longitude (°) | Latitude (°) | Time Span (Year) | Date Missing Rate |
|------|---------------|--------------|------------------|-------------------|
| ALBH | −123.49 | 48.39 | 2000–2022 | 0.61% |
| BURN | −117.84 | 42.78 | 2000–2022 | 1.27% |
| CEDA | −112.86 | 40.68 | 2000–2022 | 2.74% |
| FOOT | −113.81 | 39.37 | 2000–2022 | 3.40% |
| GOBS | −120.81 | 45.84 | 2000–2022 | 3.65% |
| RHCL | −118.03 | 34.02 | 2000–2022 | 1.79% |
| SEDR | −122.22 | 48.52 | 2000–2022 | 0.49% |
| SMEL | −112.84 | 39.43 | 2000–2022 | 0.79% |

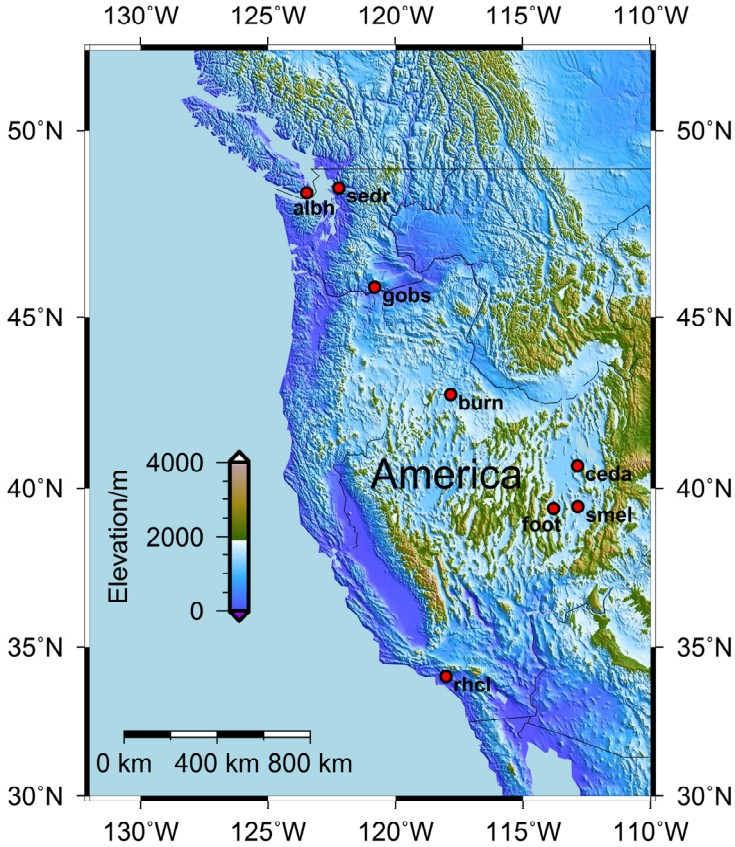

**Figure 3.** Distribution map of each GNSS station.

In order to reduce the impact of missing data on the estimation and prediction results of the noise model, the following principles were followed in the selection process of the station: (1) the selected station coordinate time series must contain data from 2000 to 2022 to ensure the consistency of the experiment and obtain reliable velocity parameter estimation; (2) in the time range from 2000 to 2022, the average missing rate of the selected station data should not exceed 5% to ensure the reliability of the predicted experiment; (3) in order to reduce the impact of inter-regional correlation on the repeatability of speed parameters and noise modeling, the selected sites should be evenly distributed.

### 3.2. Data Preprocessing

For data preprocessing, this study employed the Hector software to remove outliers and detect step discontinuities in the raw data [84,85]. After identifying the step discontinuities, they were corrected using the least squares fitting method. The corrected data were then subjected to interpolation using the Regularized Expectation Maximization (RegEM) algorithm [86,87]. This method combines the Expectation Maximization (EM) algorithm

with regularization techniques to simultaneously maximize the likelihood function and consider the smoothness of the model and noise reduction. It can effectively handle the interpolation problem of missing data [88,89]. Due to space limitations, only the comparison of interpolation results for the GBOS station with the highest missing rate in the E, N, and U components is shown in Figure 4.

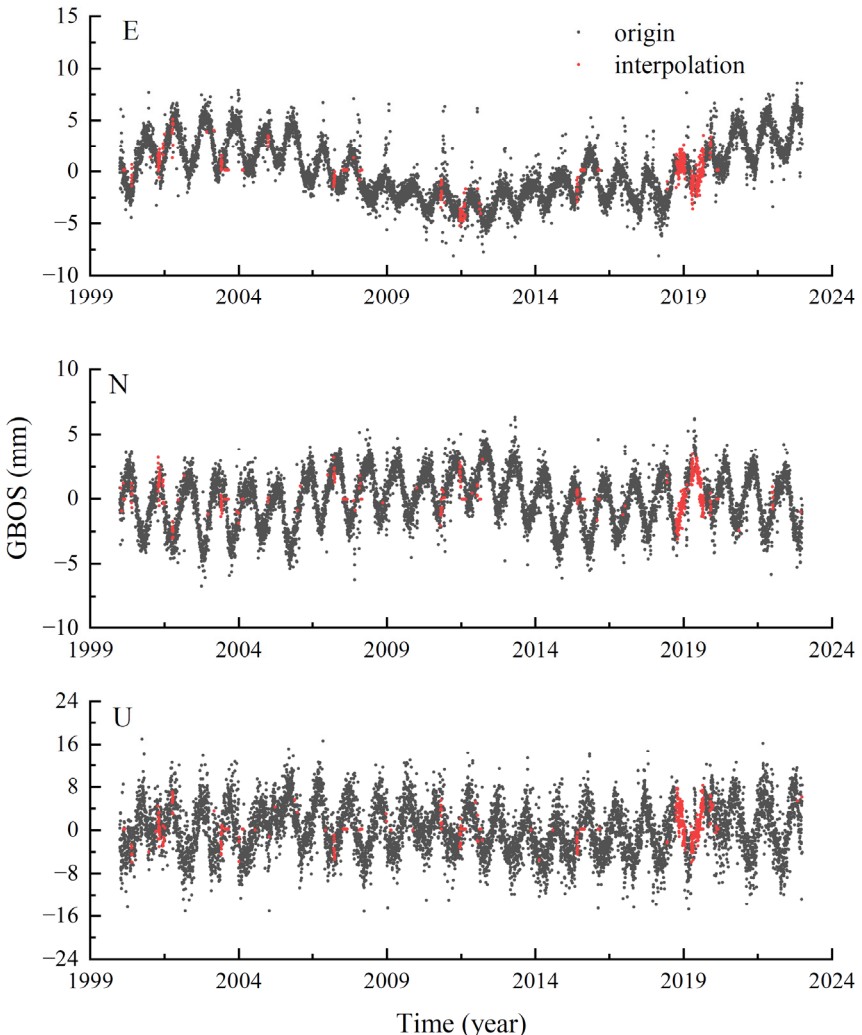

**Figure 4.** Three-direction interpolation comparison chart of GBOS station.

As shown in the figure, it can be observed that the RegEM method not only produces good interpolation results for scattered missing data but also maintains the trend of the sequence well in the presence of many continuous missing data. It successfully overcomes the limitation of the poor interpolation performance of linear interpolation at locations with continuous missing data. Moreover, it provides high-quality continuous time series data for subsequent experiments.

### 3.3. VMD Parameter Discussion

When performing data decomposition using VMD, the selection of an appropriate number of mode components *K* is crucial for achieving high-quality decomposition results in VMD. An excessively large *K* may lead to over-decomposition, while a small *K* may result in the under-decomposition of the data. To determine the optimal *K* value for the E, N, and U time series of the different stations, this study adopts the method of comparing the signal-to-noise ratio (SNR) of the decomposed data to evaluate the quality of the decomposition results. A higher SNR indicates clearer signal decomposition and a better denoising effect.

Through extensive experiments, and based on empirical rules, this study restricts the *K* value to a range of 2 to 10 and selects the *K* value within this range that yields the highest SNR as the optimal *K* value for each time series [90,91]. The definition of SNR is given as follows:

$$\text{SNR} = 10\lg\frac{\sum\limits_{i=1}^{N} f^2(i)}{\sum\limits_{i=1}^{N} \left[f(i) - g(i)\right]^2} \tag{11}$$

where $f(i)$ represents the original signal, and $g(i)$ represents the reconstructed signal. The determination of the penalty factor $\alpha$ also has a certain impact on the decomposition results in VMD; moreover, considering that selecting a penalty factor of approximately 1.5 times the decomposed data is optimal [92], in order to ensure experimental consistency, a penalty factor of 10,000 was set for all the decomposition processes in this study. The results of *K* value selection in three directions at each site are shown in Table 2.

**Table 2.** Results of *K* value selection in three directions at each site.

| Site | Direction | | |
|------|:---:|:---:|:---:|
| | **N** | **E** | **U** |
| ALBH | 3 | 6 | 3 |
| BURN | 4 | 4 | 3 |
| CEDA | 4 | 4 | 3 |
| FOOT | 3 | 8 | 5 |
| GOBS | 3 | 6 | 5 |
| RHCL | 7 | 3 | 3 |
| SEDR | 3 | 5 | 7 |
| SMEL | 7 | 3 | 5 |

## 4. Experimental Results and Analysis

### 4.1. DVMD-LSTM Prediction Results Analysis

To ensure experimental fairness and consistency, all deep learning models used in this paper are consistently divided into data sets, which are divided into training sets (2000.0 to 2011.9), validation sets (2012.0 to 2014.9), and test sets (2015.0 to 2022.9). The training set was used to train the model parameters and learn the data features. The validation set was used to fine-tune the model's hyperparameters and evaluate its performance. The test set was used for the final evaluation of the model's performance to assess its effectiveness in practical applications. The purpose of this dataset partitioning scheme was to ensure that the model had sufficient training data to fully learn the data features. Additionally, by obtaining sufficient prediction results on the test set, the optimal noise model for prediction accuracy could be evaluated.

In order to visually demonstrate the differences in the prediction results between the DVMD-LSTM model and the VMD-LSTM model, this study compares and discusses the prediction results of the decomposed IMF and residual terms using the two hybrid models. Due to space limitations, this paper only presents the prediction results of the IMF and residual terms in the U direction at the SEDR station. For detailed information, please refer to Figure 5.

From Figure 5, it can be observed that both the VMD-LSTM and DVMD-LSTM models yield good prediction results for each IMF component. However, due to the lack of apparent regularity in the residual terms, the VMD-LSTM model struggles to capture their fluctuation characteristics effectively, resulting in lower prediction accuracy and subsequently affecting the overall prediction performance of the VMD-LSTM model. To address this issue, the proposed DVMD-LSTM model conducts a secondary VMD decomposition on the residual terms obtained after the first VMD decomposition, further extracting the fluctuation information within the residual terms and significantly improving the prediction accuracy. In order to investigate whether performing multiple VMD decompositions

can further enhance accuracy, analyses were conducted on the residual terms after the second decomposition. It was found that they lack noticeable fluctuation characteristics. When these results are incorporated into the model for prediction, there is no significant improvement observed; moreover, some stations even exhibit a decrease in prediction accuracy. This indicates that increasing the number of decompositions on the residual terms may not necessarily enhance the prediction accuracy of the model. Therefore, in this study, the data after the secondary VMD decomposition were used as the feature input for the subsequent deep learning experiments.

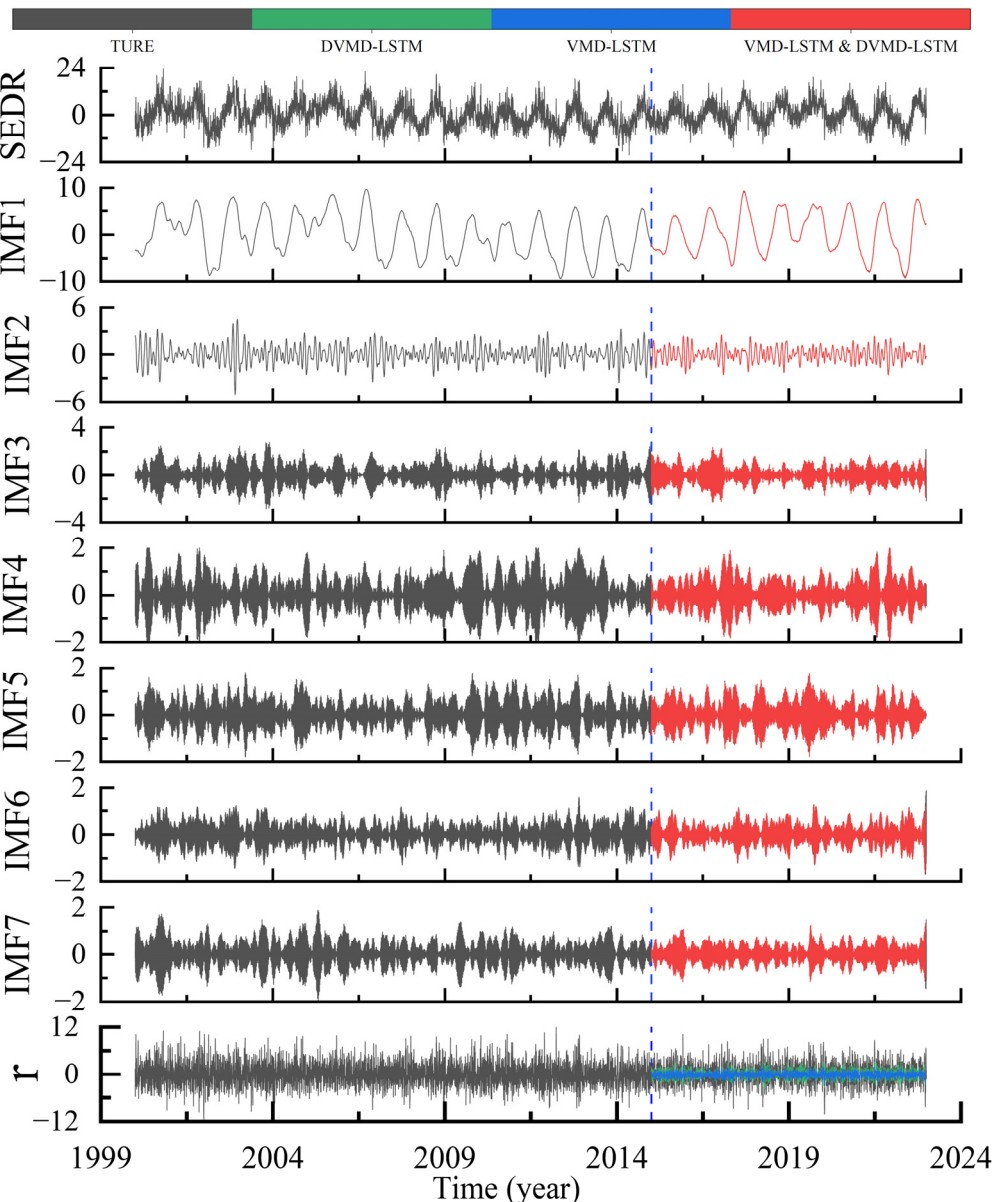

**Figure 5.** Prediction results of each IMF and residual terms under different models after VMD decomposition in the U direction of the SEDR station (the black curve represents the original data as well as the IMF components and residual terms obtained from VMD decomposition. The red curve represents the prediction results of IMF components using the DFVMD-LSTM and VMD-LSTM models, the blue curve represents the prediction results of residual terms using the VMD-LSTM model, and the green curve represents the prediction results of residual terms using the DVMD-LSTM model).

### 4.2. DVMD-LSTM Model Prediction Results and Precision Analysis

To compare the improvement in the predictive accuracy of the DVMD-LSTM model and the VMD-LSTM model compared to the LSTM model under different fluctuation amplitudes, this study conducted experiments using datasets from different stations in three directions. To better distinguish the prediction results, this study analyzed the prediction error R, which is the difference between the true values and the predicted results. Due to space limitations, this section only presents the prediction results of the SEDR station in three directions for different models, as shown in Figure 6.

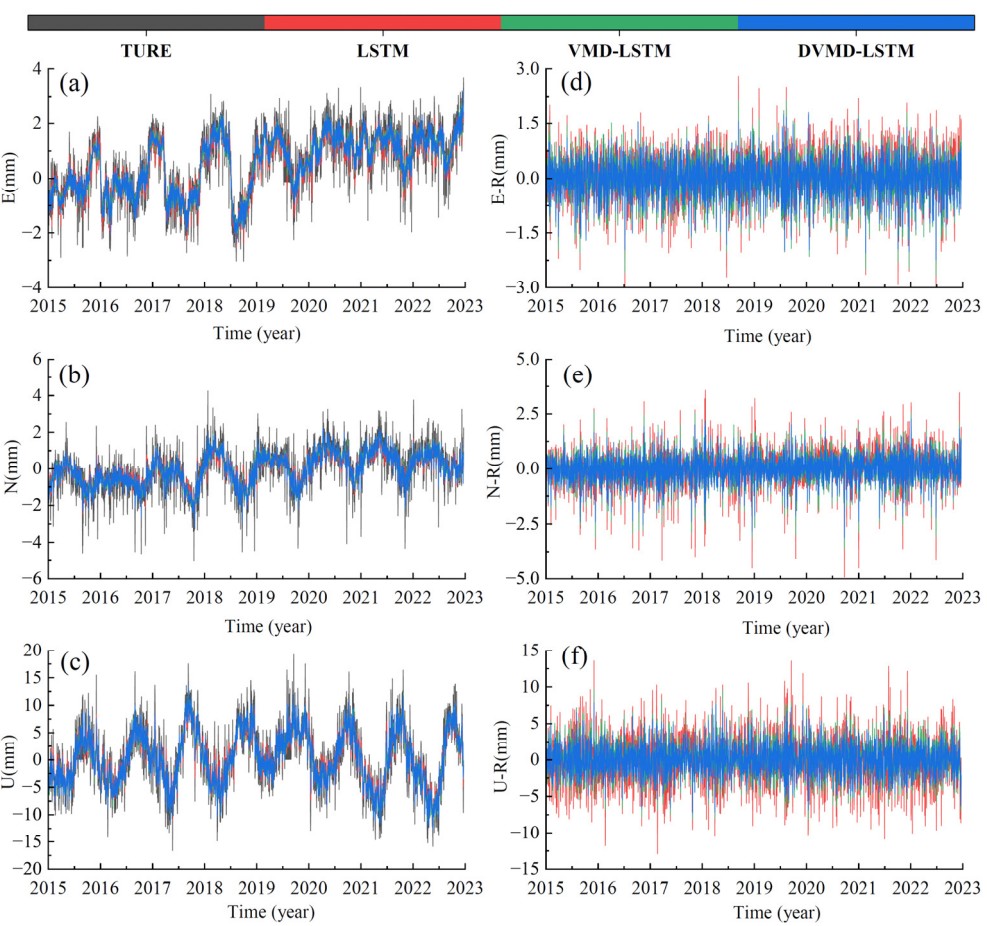

**Figure 6.** Comparison of prediction results and prediction error R in three directions of the SEDR station under different models (sub-figures (**a**–**c**) are the prediction results of each model and sub-figures (**d**–**f**) are comparison diagrams of the prediction error R of each model).

From Figure 6, it can be observed that, as the fluctuation amplitude of the original data increases, the prediction errors of different models also increase to varying degrees, with the largest errors being observed in the U direction. Compared to the LSTM model, the VMD-LSTM hybrid model better captures the fluctuation trends and amplitudes of the true values in the data and exhibits smaller variations and extremities in the prediction error R. This indicates that, after VMD decomposition, the VMD-LSTM model can capture the inherent fluctuation characteristics of the initial data more effectively, leading to more accurate predictions. Both the VMD-LSTM and DVMD-LSTM models exhibit similar prediction fluctuations and trends; however, the DVMD-LSTM model has smaller prediction errors R. This suggests that the DVMD-LSTM model not only retains the advantages of the VMD-LSTM model in predicting fluctuation trends and amplitudes but that it also achieves higher prediction accuracy.

To analyze the applicability and robustness of the DVMD-LSTM model, this study conducted predictions using the LSTM, VMD-LSTM, and DVMD-LSTM models in the E,

N, and U directions for each GNSS station. The prediction accuracy and improvement achieved by each model are summarized in Table 3, where "*I*" represents the degree of accuracy improvement of the hybrid model compared with the single LSTM model under different accuracy indexes.

**Table 3.** Comparison of the prediction results of each GNSS station in the three directions of E, N, and U under different models (the units of RMSE and MAE in the table are in mm).

| Site | ENU | LSTM | | | VMD-LSTM | | | | | | DVMD-LSTM | | | | | |
|------|-----|------|------|------|------|------|------|------|------|------|------|------|------|------|------|------|
| | | *RMSE* | *MAE* | *R2* | *RMSE* | *I*/% | *MAE* | *I*/% | *R2* | *I*/% | *RMSE* | *I*/% | *MAE* | *I*/% | *R2* | *I*/% |
| ALBH | | 0.89 | 0.65 | 0.65 | 0.76 | 13.91 | 0.55 | 14.03 | 0.74 | 13.75 | 0.67 | 24.56 | 0.49 | 24.31 | 0.80 | 22.89 |
| BURN | | 1.40 | 1.10 | 0.51 | 1.16 | 17.00 | 0.92 | 16.70 | 0.66 | 30.37 | 1.02 | 27.00 | 0.82 | 25.78 | 0.74 | 45.61 |
| CEDA | | 1.73 | 1.35 | 0.70 | 1.37 | 20.75 | 1.06 | 21.18 | 0.81 | 16.00 | 1.21 | 29.82 | 0.94 | 30.32 | 0.85 | 21.83 |
| FOOT | E | 0.58 | 0.44 | 0.13 | 0.51 | 12.91 | 0.38 | 13.51 | 0.34 | 157.6 | 0.45 | 22.12 | 0.34 | 22.27 | 0.47 | 256.7 |
| GOBS | | 1.00 | 0.70 | 0.86 | 0.86 | 13.74 | 0.58 | 16.08 | 0.90 | 4.10 | 0.77 | 23.53 | 0.52 | 24.50 | 0.92 | 6.66 |
| RHCL | | 1.62 | 1.28 | 0.61 | 1.07 | 34.08 | 0.83 | 34.78 | 0.83 | 35.51 | 0.94 | 41.63 | 0.74 | 41.91 | 0.87 | 41.40 |
| SEDR | | 0.68 | 0.53 | 0.66 | 0.58 | 15.00 | 0.45 | 15.13 | 0.76 | 14.23 | 0.50 | 27.07 | 0.39 | 26.76 | 0.82 | 24.00 |
| SMEL | | 0.57 | 0.44 | 0.40 | 0.40 | 30.80 | 0.30 | 31.08 | 0.71 | 77.69 | 0.34 | 40.11 | 0.26 | 39.98 | 0.79 | 95.60 |
| ALBH | | 0.73 | 0.57 | 0.62 | 0.55 | 24.53 | 0.43 | 24.23 | 0.78 | 26.18 | 0.49 | 32.77 | 0.38 | 32.53 | 0.83 | 33.33 |
| BURN | | 1.39 | 1.11 | 0.55 | 1.07 | 22.74 | 0.85 | 23.37 | 0.73 | 32.59 | 0.95 | 31.65 | 0.76 | 32.13 | 0.79 | 43.08 |
| CEDA | | 1.38 | 1.10 | 0.46 | 1.05 | 23.54 | 0.83 | 24.05 | 0.68 | 48.72 | 0.90 | 34.50 | 0.72 | 34.33 | 0.77 | 66.97 |
| FOOT | N | 0.59 | 0.43 | 0.48 | 0.39 | 33.45 | 0.29 | 31.81 | 0.77 | 59.65 | 0.34 | 41.35 | 0.26 | 39.95 | 0.82 | 70.25 |
| GOBS | | 0.86 | 0.63 | 0.78 | 0.63 | 26.95 | 0.46 | 26.60 | 0.88 | 13.33 | 0.56 | 34.86 | 0.41 | 34.10 | 0.91 | 16.46 |
| RHCL | | 3.14 | 2.54 | 0.46 | 1.71 | 45.59 | 1.31 | 48.53 | 0.84 | 81.39 | 1.58 | 49.55 | 1.21 | 52.28 | 0.86 | 86.19 |
| SEDR | | 0.85 | 0.63 | 0.44 | 0.66 | 22.23 | 0.50 | 21.79 | 0.66 | 50.49 | 0.56 | 34.15 | 0.42 | 33.10 | 0.76 | 72.34 |
| SMEL | | 0.55 | 0.42 | 0.45 | 0.47 | 15.62 | 0.35 | 16.54 | 0.61 | 35.42 | 0.41 | 26.53 | 0.30 | 26.91 | 0.70 | 56.60 |
| ALBH | | 3.38 | 2.60 | 0.58 | 2.89 | 14.57 | 2.25 | 13.77 | 0.69 | 19.40 | 2.51 | 25.74 | 1.96 | 0.83 | 0.77 | 32.21 |
| BURN | | 2.30 | 1.78 | 0.53 | 1.94 | 15.78 | 1.49 | 16.29 | 0.66 | 26.08 | 1.66 | 27.82 | 1.29 | 0.79 | 0.75 | 42.98 |
| CEDA | | 2.65 | 2.03 | 0.51 | 2.27 | 14.48 | 1.73 | 15.08 | 0.64 | 25.63 | 1.96 | 26.09 | 1.49 | 0.77 | 0.73 | 43.28 |
| FOOT | U | 2.39 | 1.83 | 0.31 | 1.87 | 21.89 | 1.43 | 22.23 | 0.58 | 88.11 | 1.60 | 32.94 | 1.23 | 0.82 | 0.69 | 124.3 |
| GOBS | | 2.92 | 2.22 | 0.62 | 2.28 | 22.17 | 1.72 | 22.48 | 0.77 | 24.56 | 1.99 | 32.04 | 1.53 | 0.91 | 0.82 | 33.52 |
| RHCL | | 2.45 | 1.90 | 0.31 | 2.10 | 14.50 | 1.63 | 14.04 | 0.49 | 60.46 | 1.87 | 23.68 | 1.46 | 0.86 | 0.60 | 93.85 |
| SEDR | | 3.33 | 2.62 | 0.65 | 2.37 | 28.68 | 1.87 | 28.79 | 0.82 | 26.63 | 1.96 | 41.19 | 1.54 | 0.76 | 0.88 | 35.44 |
| SMEL | | 2.36 | 1.87 | 0.32 | 1.84 | 22.38 | 1.43 | 23.12 | 0.59 | 85.49 | 1.58 | 33.17 | 1.24 | 0.70 | 0.70 | 118.9 |

From the results in Table 3, it can be observed that the VMD-LSTM model exhibits an average reduction of 19.77% in RMSE for the E direction, an average reduction of 26.83% in RMSE for the N direction, and an average reduction of 19.31% in RMSE for the U direction, outperforming the LSTM model. The VMD-LSTM model demonstrates an average reduction of 20.31% in MAE for the E direction, an average reduction of 27.12% in MAE for the N direction, and an average reduction of 19.48% in MAE for the U direction. Additionally, the VMD-LSTM model shows an average increase of 43.66% in *R2* for the E direction, an average increase of 43.47% in *R2* for the N direction, and an average increase of 44.54% in *R2* for the U direction. The experimental results indicate that the VMD-LSTM model significantly improves prediction accuracy compared to the standalone LSTM model. Although there are varying degrees of improvement in *R2*, they are observed to different degrees at different stations. Notably, the improvement is more prominent in stations where the LSTM model had lower *R2* values, suggesting that the VMD-LSTM model exhibits better explanatory power and produces predictions that closely match the observed values with improved fitting results.

Compared to the VMD-LSTM model, the DVMD-LSTM model demonstrates an average reduction of 9.71% in RMSE for the E direction, an average reduction of 8.84% in RMSE for the N direction, and an average reduction of 11.02% in RMSE for the U direction. The DVMD-LSTM model exhibits an average reduction of 9.17% in MAE for the E direction, an average reduction of 8.55% in MAE for the N direction, and an average reduction of 10.61% in MAE for the U direction. Moreover, the DVMD-LSTM model shows an average

increase of 20.68% in *R2* for the E direction, an average increase of 12.18% in *R2* for the N direction, and an average increase of 21.03% in *R2* for the U direction. The overall average *R2* value reaches 0.78, indicating a strong correlation between the DVMD-LSTM model's prediction results and the original data along with improved fitting performance. It can be concluded that the DVMD-LSTM model achieves a significant improvement in accuracy compared to the VMD-LSTM model, with particularly notable improvements in *R2*. The DVMD-LSTM model exhibits a greater improvement in the U direction, suggesting that it performs better for time series with larger fluctuations. This is because, for time series with larger fluctuations, the residual terms obtained after VMD decomposition are larger and contain more fluctuation characteristics.

In summary, the DVMD-LSTM model preserves the advantages of the VMD-LSTM model in predicting fluctuation trends and frequencies while achieving higher prediction accuracy. The results of the predictions conducted across the different directional components of various stations further validate the superiority of the proposed model. These experimental findings confirm the model's applicability and robustness, demonstrating its potential for broad utilization in the field of high-precision time series forecasting.

### 4.3. Optimal Noise Model Research

4.3.1. Comparison of Optimal Noise Models under Each Prediction Model

To further investigate whether the DVMD-LSTM model can adequately consider the noise characteristics of different datasets during the prediction process, we considered the fact that, currently, domestic and foreign scholars believe that white noise + flicker noise (FN + WN) and a small amount of random walk noise + flicker noise (RW + FN) are the optimal random models for the noise characteristics of GPS coordinate time series [93–97]. In addition, some scholars have proposed that, in GPS coordinate time series, some noise models can be represented by power law noise (PL) and the Gaussian Markov model (GGM) [98–100]. This paper takes GNSS reference stations with the same time span in North America as the research object. Four combined noise models, random walk noise + flicker noise + white noise (RW + FN + WN), flicker noise + white noise (FN + WN), power law noise + white noise (PL + WN) and Gaussian Markov + white noise (GGM + WN), were used to analyze the training set and test set data of each station. Finally, eight stations with the same optimal noise model were selected as the experimental data, and the optimal noise model of each prediction model to the prediction results of each station was calculated. The specific results are shown in Table 4.

According to Table 4, the optimal noise models differ among different stations, indicating the presence of inconsistent noise characteristics. The LSTM model exhibits significant differences between its prediction results and the optimal noise models of the original data, with an average accuracy of only 25% across all three directions. Additionally, the predominant optimal noise models are PLWN and GGMWN. This suggests that the LSTM model does not adequately consider the inherent noise characteristics of GNSS time series during prediction. In contrast, the VMD-LSTM model shows improved accuracy in capturing the optimal noise models, with an average accuracy of 42.67%. This indicates that the VMD decomposition effectively captures the noise characteristics within the IMF components; however, the noise characteristics in the residual component r are not fully captured, resulting in relatively lower overall accuracy. Therefore, the proposed DVMD-LSTM model further enhances the noise characteristics in the residual component r by performing VMD decomposition once again. As a result, the DVMD-LSTM model achieves an impressive average accuracy of 79.17% in capturing the optimal noise models. In summary, the DVMD-LSTM model adequately considers the noise characteristics of the data during the prediction process by processing the original data and decomposed residual components.

**Table 4.** The optimal noise model of each station under different models in the three directions of E, N, and U.

| Site | ENU | Optimal Noise Model | | | |
|------|-----|------|------|------|------|
| | | **TURE** | **LSTM** | **VMD-LSTM** | **DVMD-LSTM** |
| ALBH | | RW + FN + WN | PL + WN | RW + FN + WN | RW + FN + WN |
| BURN | | RW + FN + WN | PL + WN | PL + WN | RW + FN + WN |
| CEDA | | RW + FN + WN | PL + WN | PL + WN | RW + FN + WN |
| FOOT | | PL + WN | GGM + WN | FN + WN | PL + WN |
| GOBS | E | RW + FN + WN | PL + WN | RW + FN + WN | RW + FN + WN |
| RHCL | | RW + FN + WN | GGM + WN | PL + WN | RW + FN + WN |
| SEDR | | RW + FN + WN | PL + WN | PL + WN | RW + FN + WN |
| SMEL | | FN + WN | PL + WN | FN + WN | FN + WN |
| ALBH | | RW + FN + WN | PL + WN | RW + FN + WN | RW + FN + WN |
| BURN | | FN + WN | PL + WN | PL + WN | PL + WN |
| CEDA | | RW + FN + WN | PL + WN | PL + WN | RW + FN + WN |
| FOOT | | FN + WN | GGM + WN | FN + WN | FN + WN |
| GOBS | N | RW + FN + WN | PL + WN | RW + FN + WN | RW + FN + WN |
| RHCL | | RW + FN + WN | RW + FN + WN | PL + WN | PL + WN |
| SEDR | | FN + WN | GGM + WN | RW + FN + WN | FN + WN |
| SMEL | | FN + WN | PL + WN | FN + WN | FN + WN |
| ALBH | | PL + WN | PL + WN | RW + FN + WN | FN + WN |
| BURN | | PL + WN | GGM + WN | PL + WN | PL + WN |
| CEDA | | PL + WN | PL + WN | RW + FN + WN | PL + WN |
| FOOT | | PL + WN | PL + WN | FN + WN | FN + WN |
| GOBS | U | PL + WN | GGM + WN | PL + WN | FN + WN |
| RHCL | | FN + WN | PL + WN | RW + FN + WN | FN + WN |
| SEDR | | PL + WN | PL + WN | PL + WN | PL + WN |
| SMEL | | PL + WN | PL + WN | FN + WN | PL + WN |

### 4.3.2. Velocity Estimation Impact Analysis

To further investigate the quality of the prediction results from each deep learning model, this study first utilized these models to predict the original data. The optimal noise model and corresponding velocities were computed for each model's prediction results. Subsequently, these velocities were compared with the velocities obtained by calculating the optimal noise model of the original data using the Hector software [84,85]. By calculating the absolute error between the prediction results of each model and the original velocities at different measurement stations, the average absolute error between the velocities computed from the prediction results of each deep learning model and the velocities from the original data could be obtained. Finally, by comparing the average absolute error between the prediction results of each deep learning model and the velocities from the original data, the quality of the model's prediction results could be assessed. The velocities computed from the prediction results of each deep learning model under the optimal noise model at different measurement stations are shown in Table 5.

According to Table 5, in the E direction of each station, the average absolute error between the velocities predicted by the LSTM model and the velocities of the original data is 0.068 mm/year. In the N direction, it is 0.093 mm/year; in the U direction, it is 0.078 mm/year. For the VMD-LSTM model, the average absolute error between the predicted velocities and the velocities of the original data is 0.031 mm/year in the E direction, 0.060 mm/year in the N direction, and 0.060 mm/year in the U direction. As for the DVMD-LSTM model, the average absolute error between the predicted velocities and the velocities of the original data is 0.016 mm/year in the E direction, 0.042 mm/year in the N direction, and 0.047 mm/year in the U direction. Compared to the LSTM model, the VMD-LSTM model shows an average improvement of 37.67% in velocity prediction accuracy, while the DVMD-LSTM model demonstrates an average improvement of 56.80%. Compared with VMD-LSTM, the velocity prediction accuracy of the DVMD-LSTM model

is improved by 33.02% on average. Thus, both the VMD-LSTM and DVMD-LSTM models exhibit improved velocity prediction accuracy compared to the LSTM model, with the DVMD-LSTM model showing a greater improvement, further demonstrating its outstanding predictive performance.

**Table 5.** Velocity values obtained by each station under the optimal noise model.

| Site | ENU | Trend (mm/Year) | | | |
|------|-----|------|------|----------|-----------|
|      |     | TURE | LSTM | VMD-LSTM | DVMD-LSTM |
| ALBH |     | −0.041 | 0.020 | 0.055 | −0.044 |
| BURN |     | −0.108 | −0.005 | −0.051 | −0.116 |
| CEDA |     | −0.726 | −0.528 | −0.693 | −0.736 |
| FOOT | E   | 0.02 | 0.015 | 0.001 | 0.009 |
| GOBS |     | 0.659 | 0.656 | 0.672 | 0.682 |
| RHCL |     | 0.811 | 0.666 | 0.805 | 0.783 |
| SEDR |     | 0.354 | 0.341 | 0.378 | 0.313 |
| SMEL |     | 0.026 | 0.009 | 0.023 | 0.021 |
| ALBH |     | 0.327 | 0.245 | 0.276 | 0.295 |
| BURN |     | 0.124 | 0.080 | 0.116 | 0.130 |
| CEDA |     | −0.065 | −0.041 | −0.227 | −0.042 |
| FOOT | N   | 0.009 | 0.029 | −0.036 | 0.005 |
| GOBS |     | 0.063 | 0.078 | 0.029 | −0.020 |
| RHCL |     | 1.253 | 0.743 | 1.132 | 1.071 |
| SEDR |     | 0.199 | 0.170 | 0.212 | 0.195 |
| SMEL |     | 0.020 | −0.001 | −0.025 | 0.017 |
| ALBH |     | 0.383 | 0.204 | 0.131 | 0.268 |
| BURN |     | 0.241 | 0.144 | 0.238 | 0.216 |
| CEDA |     | 0.016 | 0.159 | 0.074 | 0.137 |
| FOOT | U   | 0.194 | 0.125 | 0.194 | 0.202 |
| GOBS |     | 0.301 | 0.278 | 0.283 | 0.262 |
| RHCL |     | 0.298 | 0.206 | 0.367 | 0.264 |
| SEDR |     | 0.017 | 0.022 | 0.082 | 0.04 |
| SMEL |     | 0.195 | 0.182 | 0.206 | 0.183 |

In summary, this study evaluated the performance of various prediction models by analyzing their prediction accuracy, optimal noise models, and velocity results. The results indicate that the DVMD-LSTM model outperforms the others in multiple aspects, highlighting its potential for widely applicable high-precision time series prediction with multiple noise characteristics.

## 5. Conclusions

Addressing the limitations of low prediction accuracy and inadequate consideration of noise characteristics in the VMD-LSTM model for time series forecasting, this paper proposes a high-precision GNSS time series prediction method based on DVMD and LSTM. The proposed method is comprehensively validated and tested on the daily time series data from eight North American regional GNSS stations, spanning the period from 2000 to 2022, in the E, N, and U directions. The experimental results demonstrate the following:

(1) The VMD-LSTM model shows good prediction results for each IMF value after VMD decomposition but performs poorly in predicting the residual component. The proposed DVMD-LSTM model utilizes VMD decomposition to extract the fluctuation characteristics of the residual component, leading to a significant improvement in the prediction accuracy of the residual component and enhancing the overall prediction accuracy;

(2) Compared to the initial VMD-LSTM hybrid model, the DVMD-LSTM model exhibits significant improvements in prediction accuracy. The *RMSE* values for the DVMD-LSTM model are reduced by an average of 9.71% in the E direction, 8.84% in the N

direction, and 11.02% in the U direction. Additionally, the *MAE* values decreased by an average of 9.17% in the E direction, 8.55% in the N direction, and 10.61% in the U direction. Moreover, the DVMD-LSTM model shows an average increase of 20.68% in *R2* for the E direction, an average increase of 12.18% in *R2* for the N direction, and an average increase of 21.03% in *R2* for the U direction. Across all measurement stations, the DVMD-LSTM model consistently outperforms the VMD-LSTM model, indicating its superior predictive accuracy, adaptability, and robustness;

(3)  Compared to the LSTM model, the DVMD-LSTM model achieves an average improvement of 36.50% in the accuracy of the average optimal noise model across all stations, reaching an overall accuracy of 79.17%. This demonstrates that the DVMD-LSTM model adequately considers the noise characteristics of the data during the prediction process and achieves superior prediction results. By calculating the velocities obtained from the optimal noise models, it is evident that the DVMD-LSTM model achieves an average improvement of 33.02% in velocity prediction accuracy compared to the VMD-LSTM model, further confirming the outstanding predictive performance of the DVMD-LSTM model.

**Author Contributions:** H.C. and J.H., writing—original draft preparation; X.H., T.L., K.Y. and X.M., methodology and reviewed and edited the manuscript; X.S. and Z.H., data processing and figure plotting. All authors have read and agreed to the published version of the manuscript.

**Funding:** This work was sponsored by National Natural Science Foundation of China (42104023), Major Discipline Academic and Technical Leaders Training Program of Jiangxi Province (20225BCJ23014), Jiangxi University of Science and Technology Postgraduate Education Teaching Reform Research Project (YJG2022006). Hebei Water Conservancy Research Plan (2021-27).

**Data Availability Statement:** The processing of GNSS data can be obtained at http://garner.ucsd.edu/pub/measuresESESES_products/Timeseries/Global/.

**Conflicts of Interest:** The authors declare no conflict of interest.

## Appendix A

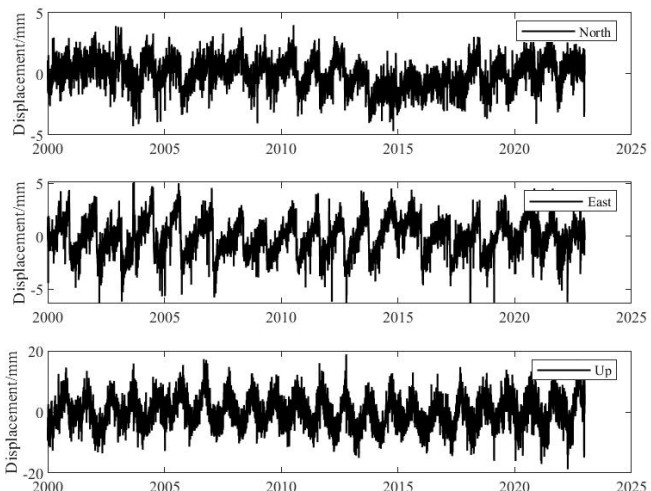

**Figure A1.** ALBH station data distribution.

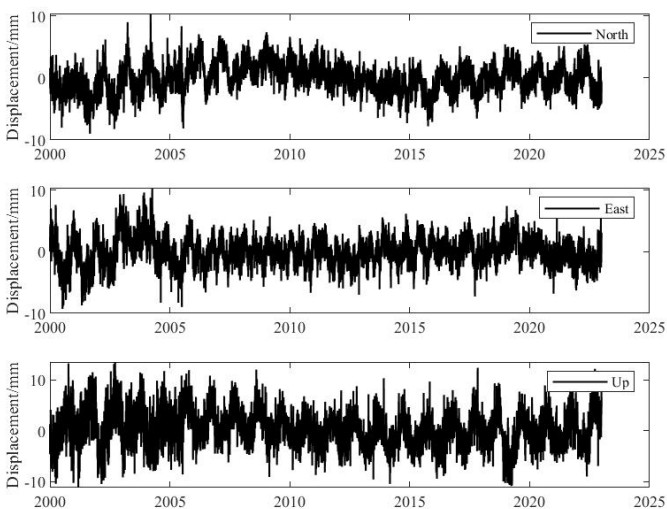

**Figure A2.** BURN station data distribution.

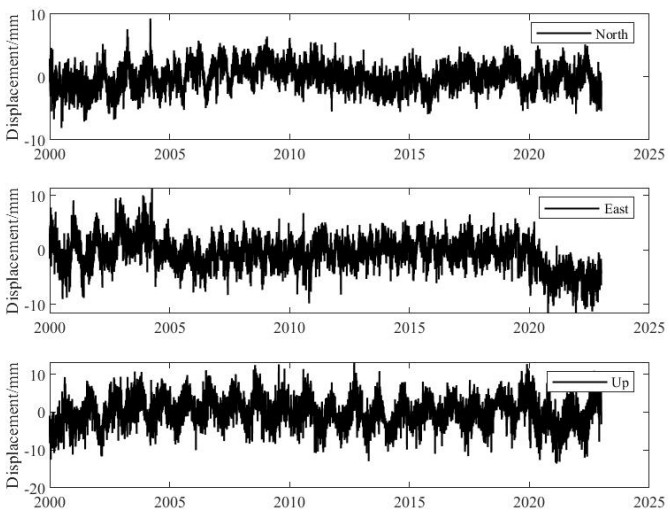

**Figure A3.** CEDA station data distribution.

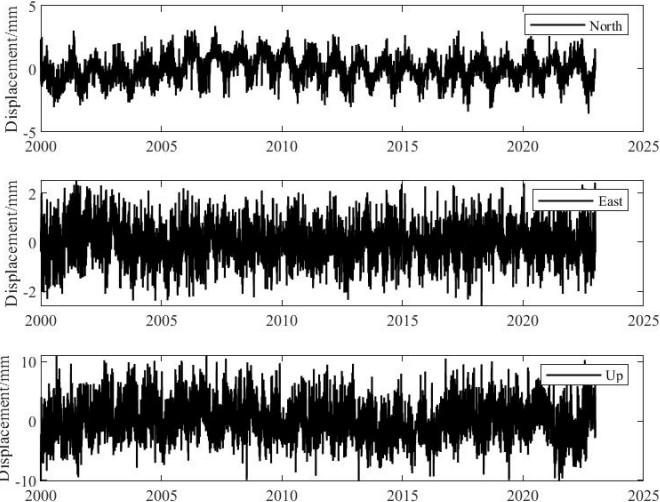

**Figure A4.** FOOT station data distribution.

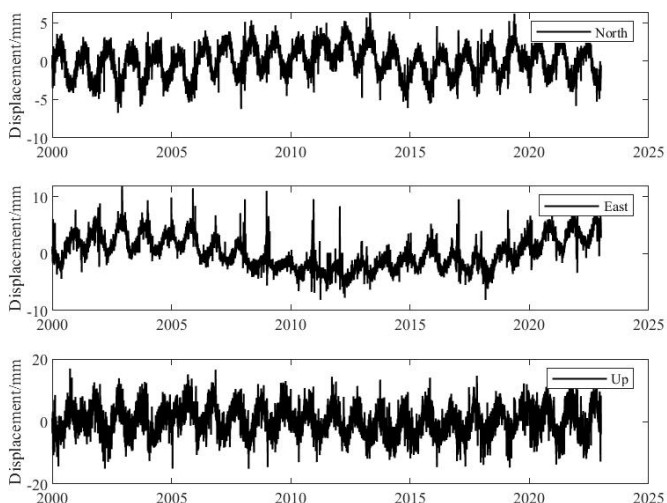

**Figure A5.** GOBS station data distribution.

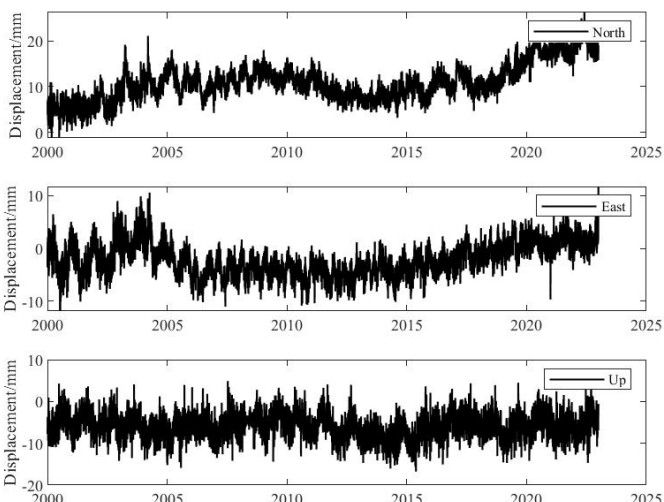

**Figure A6.** RHCL station data distribution.

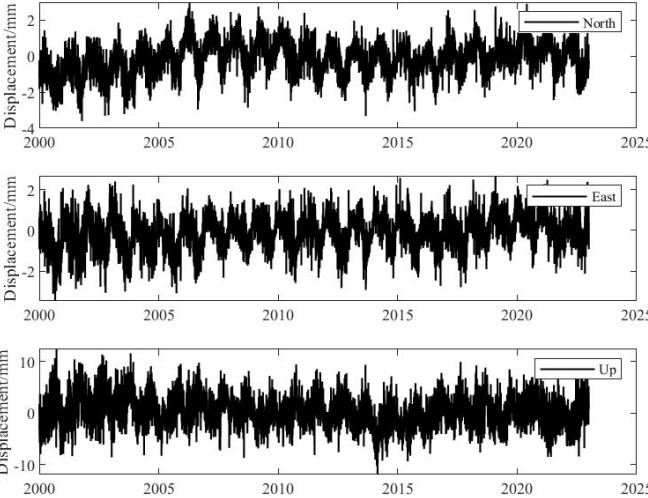

**Figure A7.** SEDR station data distribution.

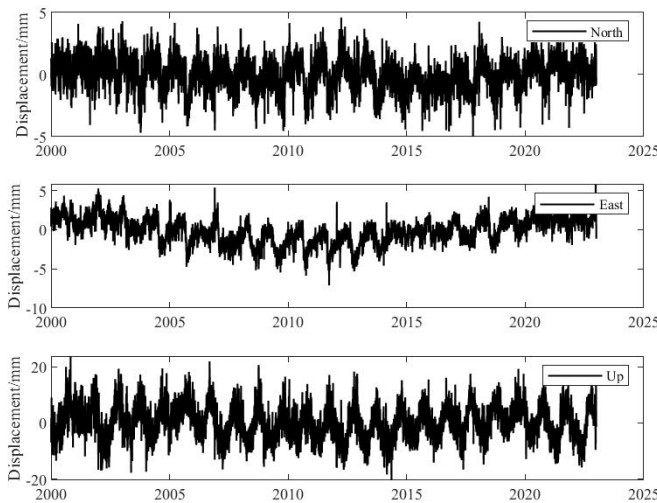

**Figure A8.** SMEL station data distribution.

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
