# Peer review of "An Improved VMD-LSTM Model for Time-Varying GNSS Time Series Prediction with Temporally Correlated Noise"

_remotesensing, doi:10.3390/rs15143694_

Round 1
Reviewer 1 Report
In the manuscript titled "Improved VMD-LSTM Model for Predicting Time-Varying GNSS Time Series with Time Dependent Noise", the DVMD-LSTM model was proposed and validated using daily E, N, and U-23 coordinate data recorded at multiple GNSS stations from 2000 to 2022. This study contains some interesting findings and are valuable for the understanding of GNSS time series prediction. However, the major flaws of the study are the ambiguous description of results and the vague and unspecific explanation of the research intention. My main issue when reviewing this manuscript was that structural issues and the use of interchangeable terminology meant that I have to read the manuscript a few times before I fully understood your definitions. Therefore, major revision has to be done before this manuscript could be accepted for publication in the data and experiments. I would suggest accepting it after the following major concerns are addressed:
1. Inadequate literature review: Presently, deep learning-based methods for time series data prediction have witnessed substantial research advancements in various domains, such as predicting time series data in fields like crustal plate movement, landslide occurrences, and bridge or dam deformation. However, the paper lacks comprehensive descriptions of these relevant studies. Moreover, the majority of the references cited are dated prior to 2021, indicating a deficiency in recent literature research within the past three years.
2. Insufficient dataset information: The author’s description of the dataset is overly simplistic, necessitating the inclusion of more comprehensive details in the paper. This should encompass the monitoring types of time series data at each station (e.g., crustal plate movement prediction, landslide prediction, bridge or dam deformation prediction) as well as the characteristics of data noise. Furthermore, it is essential to visually depict the temporal changes and noise characteristics of each site dataset through illustrations to reinforce the central argument of the paper. At last, the training set (from 2000 to 2011), validation set (from 2012 to 2014), and test set (from 2015 to 2022) are divided according to this ratio. Why was this specific ratio chosen?
3. The main idea of the paper is unclear: The residual components obtained from the initial VMD decomposition are further subjected to VMD decomposition in DVMD, improving prediction accuracy and efficiency. In Figure 5, there seems to be some deviation in the prediction accuracy of DVMD-LSTM. Would it be possible to perform a third decomposition of the residual components to further enhance prediction accuracy? Why was the choice made for two decompositions? Furthermore, when using VMD decomposition, how should specific parameters be selected? The paper only provides a method for selecting the number of components (k), but how should penalty factors and other parameters be set?
4.The analysis of the experimental results is too general: (1) The evaluation of accuracy should include the coefficient of determination R2 as a criterion. (2) The experimental results presented in section 3.3 of the paper do not effectively support the viewpoints expressed in this section. The interpretation of Figure 5 lacks clarity. Firstly, it is imperative to provide an explanation of the significance conveyed by the vertical and horizontal axes of Figure 5. Secondly, the results depicted in the figure, as well as the analysis of these results, fail to adequately elucidate the reliability of DVMD-LSTM. (3) The paper’s experimental results lack graphical representation, leading to a lack of clear and prominent focus in comparing different algorithms. For instance, Figure 6 suffers from the issue of blurry images, making it challenging to discern between different model data. (4) The experimental results in Table 2 cannot be explained “These findings highlight the significant improvement in prediction accuracy achieved by the DVMD-LSTM model through the modification of the treatment of the residual component.” (5) The analysis in section 4.2.1 of the paper lacks specificity and precision. It is crucial to include the corresponding noise images to provide clarity and facilitate comprehension of the information presented in Table 3. (6) In section 4.2.2 of the paper, it is essential to provide a detailed description of how the DVMD-LSTM’s performance is reflected by the speed value in the evaluation indicators.
5. Relying solely on LSTM and VMD-LSTM as comparative models presents challenges in showcasing the superiority of the two-stage decomposition approach. It is advisable to incorporate comparative results from additional deep learning models to enhance the demonstration, such as VMD-GRU, EMD-LSTM, etc.
6.The paper’s layout presents certain concerns, such as the occurrence of multiple table cross-page issues.

Author Response
Response to Reviewer 1 Comments
Manuscript ID: remote sensing- 2493526.
Title: An improved VMD-LSTM model for time-varying GNSS time series prediction with temporally correlated noise
We would like to thank Assistant Editor Mr. Stella Shen and the anonymous reviewers for providing an opportunity to revise the manuscript. The comments and suggestions of the reviewers are all valuable and very helpful. We have studied them carefully and have made revisions to improve the manuscript.
Detailed corrections are listed below point by point.
- Inadequate literature review: Presently, deep learning-based methods for time series data prediction have witnessed substantial research advancements in various domains, such as predicting time series data in fields like crustal plate movement, landslide occurrences, and bridge or dam deformation. However, the paper lacks comprehensive descriptions of these relevant studies. Moreover, the majority of the references cited are dated prior to 2021, indicating a deficiency in recent literature research within the past three years.
Reply: Thanks for raising the issue the inadequateness on the literature. We added more references and make further discussion (marked in red) in the revised manuscript.
- Insufficient dataset information: The author’s description of the dataset is overly simplistic, necessitating the inclusion of more comprehensive details in the paper. This should encompass the monitoring types of time series data at each station (e.g., crustal plate movement prediction, landslide prediction, bridge or dam deformation prediction) as well as the characteristics of data noise. Furthermore, it is essential to visually depict the temporal changes and noise characteristics of each site dataset through illustrations to reinforce the central argument of the paper. At last, the training set (from 2000.0 to 2012.0), validation set (from 2012.0 to 2014.0), and test set (from 2015.0 to 2022.9) are divided according to this ratio. Why was this specific ratio chosen?
Reply: Corrected. We have added relevant descriptions of the experimental data (in section 3.1) and presented the time series of each data in Appendix A. As for the noise characteristics of each site, the optimal noise model in each direction is not the same, so it is elaborated in section 4.2.1 in this paper. For details, see the third column of Table 3. Regarding the partitioning of data sets, this paper first adds the conditions for selecting sites and the reasons for determining the total time span at section 3.1, and then marks the specific reasons for dividing data sets in red at section 4.1.
- The main idea of the paper is unclear: The residual components obtained from the initial VMD decomposition are further subjected to VMD decomposition in DVMD, improving prediction accuracy and efficiency. In Figure 5, there seems to be some deviation in the prediction accuracy of DVMD-LSTM. Would it be possible to perform a third decomposition of the residual components to further enhance prediction accuracy? Why was the choice made for two decompositions? Furthermore, when using VMD decomposition, how should specific parameters be selected? The paper only provides a method for selecting the number of components (k), but how should penalty factors and other parameters be set?
Reply: In Figure 5, due to the limited inherent fluctuations and patterns in the residual terms, it is not possible for the DVMD-LSTM model to perfectly match the original data, although it exhibits significantly higher accuracy compared to the VMD-LSTM model.
For “VMD decomposition”, the revised manuscript has now included discussions and experiments on multiple decompositions of the residual terms. After the second decomposition, the residual terms show no noticeable fluctuation characteristics. Furthermore, when the data obtained from the third decomposition of the residual terms is used for prediction, the results are not significantly different from the second decomposition. Additionally, there is a phenomenon of decreased prediction accuracy observed in some stations. Please refer to section 4.1 for specific modifications.
Regarding the parameter selection in VMD, the paper has added a dedicated section (section 3.3) to provide detailed explanations.
4.The analysis of the experimental results is too general: (1) The evaluation of accuracy should include the coefficient of determination R2 as a criterion.
Reply: We have incorporated the coefficient of determination (R2) as an evaluation metric and conducted a related analysis on R2. The corresponding modifications can be found in Section 4.2.
5.The experimental results presented in section 3.3 of the paper do not effectively support the viewpoints expressed in this section. The interpretation of Figure 5 lacks clarity. Firstly, it is imperative to provide an explanation of the significance conveyed by the vertical and horizontal axes of Figure 5. Secondly, the results depicted in the figure, as well as the analysis of these results, fail to adequately elucidate the reliability of DVMD-LSTM.
Reply: To improve the structure of the paper, the content originally in Section 3.3 has been divided and reorganized into Section 4.1. We also modified Figure 5 together with relevant descriptions.
6.The paper’s experimental results lack graphical representation, leading to a lack of clear and prominent focus in comparing different algorithms. For instance, Figure 6 suffers from the issue of blurry images, making it challenging to discern between different model data.
Reply: Thanks for the nice comment. Regarding the blurriness of the images in Figure 6, it was intentional to enhance the visibility of the prediction results of different models. We set the transparency of the curves to 30% while plotting them, which resulted in a visually blurred effect. Additionally, to better differentiate the prediction results, this study included the differences between the prediction results and the ground truth for comparison.
Regarding the lack of graphical representation, we have added relevant plots of the time series for each measurement station in the appendix A. However, considering the challenge of visualizing different model prediction results for each measurement station, we have randomly selected the SEDR station for description purposes.
7.The experimental results in Table 2 cannot be explained “These findings highlight the significant improvement in prediction accuracy achieved by the DVMD-LSTM model through the modification of the treatment of the residual component.”
Reply: Thanks, we checked it and modified it to: From the results in Table 3, it can be observed that the VMD-LSTM model exhibits an average reduction of 19.77% in RMSE for the E direction, an average reduction of 26.83% in RMSE for the N direction, and an average reduction of 19.31% in RMSE for the U direction, outperforming the LSTM model. The VMD-LSTM model demonstrates an average reduction of 20.31% in MAE for the E direction, an average reduction of 27.12% in MAE for the N direction, and an average reduction of 19.48% in MAE for the U di-rection. Additionally, the VMD-LSTM model shows an average increase of 43.66% in R2 for the E direction, an average increase of 43.47% in R2 for the N direction, and an average increase of 44.54% in R2 for the U direction. The experimental results indicate that the VMD-LSTM model significantly improves the prediction accuracy compared to the standalone LSTM model. Although there are varying degrees of improvement in R2, they are observed in different degrees at different stations. Notably, the improvement is more prominent in stations where the LSTM model had lower R2 values, suggesting that the VMD-LSTM model exhibits better explanatory power and produces predictions that closely match the observed values with improved fitting results.
8.The analysis in section 4.2.1 of the paper lacks specificity and precision. It is crucial to include the corresponding noise images to provide clarity and facilitate comprehension of the information presented in Table 3.
Reply: The description of the various noise models has been revised. Considering that the focus of this research is to assess the performance of deep learning prediction models, the paper only compares the optimal noise models obtained from each prediction model with the optimal noise model of the original data to determine the prediction accuracy. As for the analysis of noise models in subsequent prediction results, further improvements will be made in the subsequent experiments.
9.In section 4.2.2 of the paper, it is essential to provide a detailed description of how the DVMD-LSTM’s performance is reflected by the speed value in the evaluation indicators.
Reply: This revised manuscript now incorporated the relevant content on how the velocity values are used to evaluate the prediction results of each deep learning model. The specific details can be found in Section 4.3.2.
- Relying solely on LSTM and VMD-LSTM as comparative models presents challenges in showcasing the superiority of the two-stage decomposition approach. It is advisable to incorporate comparative results from additional deep learning models to enhance the demonstration, such as VMD-GRU, EMD-LSTM, etc.
Reply: The main purpose of this article is to highlight the predictive advantages of the DFVMD-LSTM model. Therefore, the manuscript compares it with the VMD-LSTM model and a single LSTM model through multiple sites and methods. Therefore, the manuscript does not include other mixed depth models such as VMD-GRU and EMD-LSTM for comparison. This content will be discussed through experiments in future research. In our future work, we plan to package and develop all the algorithms as an open-source software package, which will provide a valuable resource for studying and researching these models collectively.
11.The paper’s layout presents certain concerns, such as the occurrence of multiple table cross-page issues.
Reply: Thanks for the valuable comments, we have rearranged the full text to avoid the phenomenon of cross-page tables.

Reviewer 2 Report
This paper proposes a dual variational modal decomposition long short-term memory (DVMD-LSTM) model to improve the prediction accuracy of residual terms in GNSS time series prediction. From the perspective of optimal noise model research, this method has practical value and can enhance the accuracy of prediction results.
However, there are some logical issues in the manuscript that need to be clarified:
1):This study utilized the signal-to-noise ratio (SNR) method to determine the value of K for VMD decomposition at each station. The selection of the parameter K is a key focus of this research, but the specific procedure is not explained in the paper. If the residual term r obtained after the first VMD only contains Gaussian noise, the subsequent steps would have no practical significance.
2):The authors should emphasize the significance of performing VMD decomposition on the residual term r in the manuscript. In principle, when the VMD decomposition level is high, the residual term r would only contain Gaussian white noise. Performing VMD-LSTM on white noise would only increase the algorithm complexity and may not be beneficial for practical applications. This should be addressed and explained in the paper.
3):From the experimental results, it appears that the residual term r contains some fluctuation trends. The authors should clarify whether these signals are caused by incomplete VMD decomposition.
4):Different GNSS stations in different regions have different noise models. The authors only analyzed data from 8 stations, which may not be sufficient. It is recommended to supplement the results with data from additional stations for further validation.
Author Response
Response to Reviewer 2 Comments
Manuscript ID: remote sensing- 2493526.
Title: An improved VMD-LSTM model for time-varying GNSS time series prediction with temporally correlated noise
We would like to thank Assistant Editor Mr. Stella Shen and the anonymous reviewers for providing an opportunity to revise the manuscript. The comments and suggestions of the reviewers are all valuable and very helpful. We have studied them carefully and have made revisions to improve the manuscript.
Detailed corrections are listed below point by point:
1)This study utilized the signal-to-noise ratio (SNR) method to determine the value of K for VMD decomposition at each station. The selection of the parameter K is a key focus of this research, but the specific procedure is not explained in the paper. If the residual term r obtained after the first VMD only contains Gaussian noise, the subsequent steps would have no practical significance.
Reply: Section 3.3 of the revised manuscript provides an introduction and explanation of the VMD parameter selection. The results of selecting the K values for each measurement station are presented in Table 2.
Table 2. Results of K value selection in three directions at each site
Site |
Direction |
||
N |
E |
U |
|
ALBH |
3 |
6 |
3 |
BURN |
4 |
4 |
3 |
CEDA |
4 |
4 |
3 |
FOOT |
3 |
8 |
5 |
GOBS |
3 |
6 |
5 |
RHCL |
7 |
3 |
3 |
SEDR |
3 |
5 |
7 |
SMEL |
7 |
3 |
5 |
Li et al. (2017) and Li et al. (2022) have shown that the residual terms obtained after VMD decomposition not only contain Gaussian white noise, but also exhibit a certain fluctuation trend. This conclusion can be consistent in Figure 5.
Reference as follows:
[77] Li Y, Li Y, Chen X, et al. Denoising and feature extraction algorithms using NPE combined with VMD and their applications in ship-radiated noise. J. Symmetry. 2017, 9(11): 256.
[78 ] Li C, Wu Y, Lin H, et al. ECG denoising method based on an improved VMD algorithm. J. IEEE. Sens. J. 2022, 22(23): 22725-22733.
2)The authors should emphasize the significance of performing VMD decomposition on the residual term r in the manuscript. In principle, when the VMD decomposition level is high, the residual term r would only contain Gaussian white noise. Performing VMD-LSTM on white noise would only increase the algorithm complexity and may not be beneficial for practical applications. This should be addressed and explained in the paper.
Reply: In theory, if VMD can capture all frequency components of a signal and decompose them into individual modes, the residual terms should only contain white noise. In practical applications, due to limitations of the VMD algorithm or inherent characteristics of the signal, the residual terms may contain some non-white noise components, especially lots of previous studies conformed that GNSS time series is contain colored noise (Mao et al., 1999; Amiri-Simkooei et al., 2009; He et al., 2019; Santamaría‐Gómez et al., 2021). These non-white noise components could be random noise, high-frequency noise, or low-energy components that were not fully decomposed. Therefore, this paper further decomposes the residual terms obtained after the initial decomposition to mitigate the adverse effects of incomplete VMD decomposition. Experimental results demonstrate that this approach significantly improves the prediction accuracy of the model. The relevant explanation has been added to section 2.3.
- Santamaría‐Gómez, A., & Ray, J. (2021). Chameleonic noise in GPS position time series. Journal of Geophysical Research: Solid Earth, 126(3), e2020JB019541.
- Amiri-Simkooei, A. R. (2009). Noise in multivariate GPS position time-series. Journal of Geodesy, 83, 175-187.
- Mao, A., Harrison, C. G., & Dixon, T. H. (1999). Noise in GPS coordinate time series. Journal of Geophysical Research: Solid Earth, 104(B2), 2797-2816.
- He, X., Bos, M. S., Montillet, J. P., & Fernandes, R. M. S. (2019). Investigation of the noise properties at low frequencies in long GNSS time series. Journal of Geodesy, 93(9), 1271-1282.
3)From the experimental results, it appears that the residual term r contains some fluctuation trends. The authors should clarify whether these signals are caused by incomplete VMD decomposition.
Reply: In theory, VMD decomposes data to produce IMF1, IMF2, IMF3…, IMFn and r with no fluctuation trend. However, in practical applications, VMD decomposition is affected by algorithm limitations and inherent characteristics of the signal. Attempting to eliminate the fluctuation characteristics in the residual term may lead to excessive decomposition of VMD, leading to a decrease in prediction accuracy. Therefore, this article uses the signal-to-noise ratio method to determine a reasonable K-value for optimal decomposition and minimize the fluctuation characteristics in the residual term as much as possible.
4)Different GNSS stations in different regions have different noise models. The authors only analyzed data from 8 stations, which may not be sufficient. It is recommended to supplement the results with data from additional stations for further validation.
Reply: This paper adds the requirement of site selection in section 3.1. Considering that each site has three directions and the operation of deep learning model is extremely time-consuming, this paper only randomly selects 8 sites that meet the requirements for experiments. In future research, we will try to add more sites to select different conditions for extensive experiments to ensure the applicability and robustness of the algorithm.

Round 2
Reviewer 1 Report
1.accept
Reviewer 2 Report
All my concerns have been responsed well, I recommend this manuscript to be accepted in Remote Sensing.